# Expression of *Saccharomyces cerevisiae RER2* Gene Encoding *Cis*-Prenyltransferase in *Trichoderma atroviride* Increases the Activity of Secretory Hydrolases and Enhances Antimicrobial Features

**DOI:** 10.3390/jof9010038

**Published:** 2022-12-26

**Authors:** Urszula Perlińska-Lenart, Sebastian Graczyk, Sebastian Piłsyk, Jacek Lenart, Agata Lipko, Ewa Swiezewska, Przemysław Bernat, Joanna S. Kruszewska

**Affiliations:** 1Institute of Biochemistry and Biophysics, Polish Academy of Sciences, Pawińskiego 5a, 02-106 Warsaw, Poland; 2Mossakowski Medical Research Centre, Polish Academy of Sciences, Pawińskiego 5, 02-106 Warsaw, Poland; 3Department of Industrial Microbiology and Biotechnology, Faculty of Biology and Environmental Protection, University of Lodz, Banacha Street 12/16, 90-237 Lodz, Poland

**Keywords:** *cis*-prenyltransferase, glycosylation, *T. atroviride*, antimicrobial activity

## Abstract

Some *Trichoderma* spp. exhibit natural abilities to reduce fungal diseases of plants through their mycoparasitic and antagonistic properties. In this study, we created new *Trichoderma atroviride* strains with elevated antifungal activity. This effect was achieved by improving the activity of *cis*-prenyltransferase, the main enzyme in dolichol synthesis, by expressing the *RER2* gene from *Saccharomyces cerevisiae*. Since dolichyl phosphate is the carrier of carbohydrate residues during protein glycosylation, activation of its synthesis enhanced the activities of dolichyl-dependent enzymes, DPM synthase and *N*-acetylglucosamine transferase, as well as stimulated glycosylation of secretory proteins. Cellulases secreted by the transformants revealed significantly higher levels or activities compared to the control strain. Consequently, the resulting *Trichoderma* strains were more effective against the plant pathogens *Pythium ultimum*.

## 1. Introduction

Some species of the filamentous fungus *Trichoderma* exhibit biocontrol activity against fungal plant pathogens. Preparations containing *Trichoderma* spp. are presently used commercially for biological control against plant pathogens such as *Pythium* spp., *Phytophthora* spp., *Botrytis* spp., *Rhizoctonia* and *Fusarium* spp. The antagonistic properties of *Trichoderma* are based on multiple mechanisms [1]. Biocontrol can result from a direct interaction between the pathogen and the biocontrol agent, which involves synthesis of hydrolases and metabolites acting synergistically. *Trichoderma* attached to the pathogenic fungus produces cell wall-degrading enzymes. They allow assimilation of the released carbohydrates and penetration into the hypha of the pathogen. A comparative transcriptomic analysis of three *Trichoderma* species, *T. reesei*, *T. atroviride* and *T. virens*, had shown different transcriptomic responses upon recognition before the physical contact with the alien hyphae [2]. The types of genes expressed distinguished *T. atroviride* and *T. virens* from *T. reesei*. *T. reesei* mainly expressed genes for nutrient acquisition, which could be explained by an attempt to compete for nutrients with the other fungus. *T. virens* and *T. atroviride* differ in their antifungal strategy, the former poisoning the pathogenic fungus with gliotoxin and the latter using antibiosis and hydrolytic enzymes. In addition, *T. atroviride* produces 6-pentyl-α-pyrone, a volatile compound with antifungal activity that is not formed by *T. virens* [2].

Since *T. atroviride* carries out a hydrolytic attack on the pathogen cell wall as a strategy of mycoparasitism, we attempted to improve its antifungal characteristics by enhancing its hydrolytic properties.

The hydrolytic enzymes secreted by *Trichoderma* are heavily glycosylated with both *N*- and *O*-linked carbohydrates [3], accounting for as much as 12–24% of their molecular mass [4]. The *O*-linked glycans are attached in a linker connecting the catalytic and substrate binding domains and are required for their proper relative positioning. *N*-glycosylation is connected with protein folding, and it is crucial for recognition of misfolded glycoproteins [5,6]. Any dysfunction of glycosylation results in the accumulation of affected proteins in the endoplasmic reticulum and down-regulation of their genes [7,8]. These data suggest that an enhanced glycosylation of secretory proteins in *T. atroviride* could improve their secretion. Glycoproteins would be folded correctly, more rapidly and in a higher proportion. Furthermore, such effective glycosylation could also improve the activity of the secretory enzymes by stabilizing them outside the cell as well [4,6,9,10,11].

Our previous results showed that, indeed, glycosylation processes and protein secretion in *T. reesei* were stimulated by overproduction of GDP-mannose and dolichyl phosphate mannose (DPM), *O*- and *N*-glycosylation substrates [12,13,14].

In this study, we present the effects of expression of the yeast *RER2* gene coding for *cis*-prenyltransferase (dehydrodolichyl diphosphate synthase) (Figure 1) in the biocontrol *T. atroviride* strain P1.

*Cis*-prenyltransferase, a key enzyme in dolichol synthesis, is the first enzyme of the polyprenyl branch of the mevalonate pathway. This enzyme catalyzes the elongation of polyprenyl chain by sequential addition of isopentenyl diphosphate (IPP) to farnesyl diphosphate (FPP) [15,16]. The phosphorylated form of dolichols is indispensable as a carbohydrate carrier in protein glycosylation.

Transformants expressing *RER2S.c* gene had an elevated activity of *cis*-prenyltransferase and also of the dolichol-dependent DPM synthase as well as *N*-acetylglucosamine transferase. Moreover, they showed significantly higher levels of *O*- and *N*-glycosylation of secretory proteins. The activities of hydrolytic enzymes secreted to the cultivation medium were increased, and their maximum activity correlated with maximum glycosylation. Probably as a consequence of their higher hydrolytic activity, the new strains also had substantially improved antimicrobial ability.

## 2. Materials and Methods

### 2.1. Strains and Cultivation Media

*T. atroviride* strain P1 (“*Trichoderma harzianum*” ATCC 74058) was used as the recipient strain for transformation.

*Escherichia coli* strain JM 109 was used for plasmid propagation [17].

*T. atroviride* was cultivated at 28 °C on a rotary shaker (250 r.p.m.) in 2-l shake flasks containing 1 l of minimal medium (MM): 1 g MgSO_4_ × 7H_2_O, 6 g (NH_4_)_2_SO_4_, 10 g KH_2_PO_4_, 3 g sodium citrate × 2H_2_O and trace elements (25 mg FeSO_4_ × 7H_2_O, 2.7 mg MnCl_2_ × 4H_2_O, 6.2 mg ZnSO_4_ × 7H_2_O, 14 mg CaCl_2_ × 2H_2_O) per liter and 1% (*w*/*v*) lactose or glucose as a carbon source. The flasks were inoculated with 42 × 10^6^ conidia/l medium.

*Pythium ultimum* the MUCL 16,164 from MUCL Culture Collection, Belgium, were grown on potato-dextrose agar (PDA) (BioShop, Burlington, ON, Canada).

*S. cerevisiae* strain BY4741 (Euroscarf: *MATa; his3∆ 1; leu2∆ 0; met15∆ 0; ura3∆ 0*) was used for DNA and RNA isolation. The yeast strain was grown in YPD medium (1% (*w*/*v*) yeast extract, 2% (*w*/*v*) bactopeptone and 2% (*w*/*v*) glucose).

### 2.2. Analysis of Fungal Growth

Fungal dry weight was determined by filtering culture samples through G1 sintered glass funnels, washing the biomass with water and drying to a constant weight at 110 °C.

### 2.3. Expression of the S. cerevisiae RER2 Gene in T. atroviride

*T. atroviride* P1 was cotransformed with the yeast *RER2* gene fused under the *A. nidulans gpdA* gene promoter (glyceraldehyde-3 phosphate dehydrogenase) and *trpC* (indole-3-glycerol phosphate synthase) terminator using pAN52-1NotI plasmid (NCBI Acc. No. Z32697), as described previously [18]. The pRMLex_30 plasmid with *E. coli* hygromycin B phosphotransferase gene (*hph*) fused between promoter and terminator elements of *Trichoderma pki1* (coding for pyruvate kinase) and *cbh2* (encoding cellobiohydrolase II) genes was used as a partner in cotransformation [19]. Transformants were selected for hygromycine B resistance on plates containing 1.2 M sorbitol and 200 μg/mL hygromycin B.

The transformants were then cultivated in liquid MM medium for preparation of DNA.


*Molecular biology methods*


### 2.4. Nucleic Acids Isolation

Chromosomal DNA was isolated from *T. atroviride* using the Promega Wizard Genomic DNA Purification kit. Total RNA was isolated using the single-step method described by Chomczynski and Sacchi [20]. Other molecular biology procedures were performed according to standard protocols [21].

### 2.5. RER2 Copies Integrated into T. atroviride Genome

Quantitative real-time PCR (qRT-PCR) analysis was performed to estimate the copy number of the transgenes in the *Trichoderma* genome using the relative standard curve method [22,23,24]. Serial 1:10 dilutions of standard samples containing from 300,000 to 300 copies of the *RER2* gene were prepared by mixing the wild-type *Trichoderma* genomic DNA with pGEM-*RER2* plasmid. Each reaction contained an adequate amount of DNA calculated using the online tool (DNA Copy Number and Dilution Calculator, Thermo Fisher Scientific, Warsaw, Poland), 0.5 μM of each primer (Appendix A) and 7.25 μL 3color 2xHS-qPCR Master Mix Sybr (A&A Biotechnology, Gdynia, Poland) in a total volume of 15 μL. All reactions were run on a LightCycler^®^ 96 (Roche Diagnostics GmbH, Mannheim, Germany) using the following program: 360 s 95 °C, 40 times (15 s 95 °C; 60 s 60 °C), followed by a melting curve generated from 65 °C to 90 °C. Four technical replicates were used for both the standard curve and the test samples. The single product was amplified by all primer pars tested. Raw data were processed using LightCycler^®^ 96 software Version 1.1.0.1320 (Roche Diagnostics GmbH, Mannheim, Germany), and the transgene copy number was calculated automatically.

### 2.6. Quantitative Reverse Transcription PCR (RT-qPCR)

One-step RT-qPCR assay was performed using a LightCycler^®^ EvoScript RNA SYBR^®^ Green I Master Kit (Roche Diagnostics GmbH, Mannheim, Germany) and was carried out in a 20 µL reaction, which consisted of 5 µL mRNA (25 pg), 0.4 µM each of forward and reverse primers (Appendix A) and 4 µL of Master Mix, 5× concentrated containing enzymes for reverse transcription and PCR, RT-qPCR reaction buffer; dATP, dCTP, dGTP and dUTP; Mg(OAc)_2_; SYBR^®^ Green I dye and proprietary additives. For the negative controls, mRNA was substituted with molecular grade water. The assay was performed using a LightCycler96^®^ (Roche Diagnostics GmbH, Mannheim, Germany) with an initial incubation at 60 °C for 15 min for RT followed by preincubation at 95 °C for 10 min. Forty-five cycles of amplification were performed using a thermal cycling profile of 95 °C for 10 s, 58 °C for 30 s. Subsequently, a melting curve was recorded by holding at 95 °C for 10 s, cooling to 65 °C for 60 s and then heating at 0.1 °C/s up to 97 °C. The amplification and melting curve data were collected and analyzed using the LightCycler96^®^ software 1.0.


*Biochemical methods*


### 2.7. Cell Membrane Preparation

Mycelium was harvested by filtration, washed with water and suspended in 50 mM Tris-HCl, pH 7.4, containing 15 mM MgCl_2_ and 9 mM 2-mercaptoethanol. Cells were homogenized in a beadbeater with glass beads (0.5 mm), and the homogenate was centrifuged at 5000× *g* for 10 min to remove cell debris and unbroken cells. The supernatant was centrifuged at 100,000× *g* for 1 h. The membrane pellet was homogenized in 50 mM Tris-HCl, pH 7.4, containing 3.5 mM MgCl_2_ and 6 mM β-mercaptoethanol and used as the source of enzyme. The whole procedure was performed at 4 °C [18].

### 2.8. Cis-Prenyltransferase (EC 2.5.1.20) Activity

The enzyme activity was assayed in the membrane fraction by incubation (final volume 250 µL) of 500 µg of membrane proteins with 4 µg FPP, 50 mM sodium phosphate buffer pH 7.4, 0.5 mM MgCl_2_, 20 mM 2-mercaptoethanol, 10 mM KF and 3 × 10^5^ cpm [^14^C]IPP (sp. act.: 52 Ci/mol) (ARC, St. Louis, MO, USA). After 90 min incubation at 30 °C, the reaction was terminated by addition of 4 mL of chloroform–methanol (3:2 *v*/*v*). The protein pellet was removed by centrifugation, and the supernatant was washed three times with 1/5 volume of 10 mM EDTA in 0.9% NaCl. The organic phase was concentrated under a stream of nitrogen and subjected to thin-layer chromatography on HPTLC RP-18 plates developed in 50 mM H_3_PO_4_ in acetone. The zone containing the radiolabeled polyprenols was scraped off, and the radioactivity was measured in a scintillation counter [18,25].

### 2.9. DPM Synthase (EC 2.4.1.83) Activity

DPM synthase activity was measured in the pelleted membrane fraction by incubating it with GDP [^14^C]Mannose (sp. act.: 50–60 Ci/mol) (ARC, St. Louis, MO, USA) and 5 ng of dolichyl phosphate (Dol-P) [26] according to Kruszewska et al. [27].

### 2.10. N-acetylglucosamine Transferase (EC 2.7.8.15) Activity

*N*-acetylglucosamine transferase activity was measured in the membrane fraction by incubation for 30 min at 30 °C of 200 µg of membrane proteins in a total volume of 50 µL containing 1 × 10^5^ cpm UDP [^14^C]*N*-acetylglucosamine (sp. act.: 249 Ci/mol) (ARC, St. Louis, MO, USA) and 5 ng of Dol-P in 40 mM Tris/HCl buffer pH 7.4 with 10 mM MgCl_2_ and 0.1% Nonidet P-40 [28]. The reaction was stopped by the addition of 4 mL of chloroform–methanol (3:2 *v*/*v*). Formation of radioactive dolichyl diphosphate *N*-acetylglucosamine and dolichyl diphosphate chitobiose was measured in the organic fraction in a scintillation counter.

### 2.11. FPP Synthase (EC2.5.1.10) Activity

The FPP synthase activity was measured in cell-free extract obtained from transformants and the control *T. atroviride* P1 strain. After 168 h of cultivation, mycelia of *Trichoderma* were harvested by filtration, washed with water and suspended in 50 mM phosphate buffer, pH 7.5, containing 1 mM MgCl_2_ and 5 mM iodoacetamide. Cells were homogenized in a beadbeater with glass beads (0.5 mm), and the homogenate was centrifuged at 5000× *g* for 10 min to remove cell debris and unbroken cells. The resulting supernatant was centrifuged again at 100,000× *g* for 1 h to remove the membrane pellet, and the obtained supernatant was used as the source of FPP synthase. The whole procedure was performed at 4 °C.

The FPP synthase activity was measured in 100 µL of reaction mixture containing 50 mM phosphate buffer, pH 7.5, 1 mM MgCl_2_, 5 mM iodoacetamide, 60 µM isopentenyl diphosphate (IPP), 1 × 10^5^ cpm [^14^C] IPP (sp. act.: 52 mCi/mM) (ARC, St. Louis, MO, USA), 120 µM dimethylallyldiphosphate (DMAPP) and 150 µg protein [29,30]. After a 5-min incubation at 37 °C, samples were ice-chilled, and 0.5 mL of water was added, followed by 1 mL of hexane and 0.2 mL of 1 M HCl to dephosphorylate the products. The samples were shaken for 30 min at 37 °C. The mixture was ice-chilled and mixed vigorously. The upper phase was washed three times with water and subjected to TLC on silica gel 60 plates in benzene-ethyl acetate, 7:1. Radioactive spots were localized by autoradiography, identified by co-chromatography with unlabeled standards and then scraped off, and the radioactivity was measured in a scintillation counter [29,30].

### 2.12. Squalene Synthase (EC 2.5.1.21) Activity

Squalene synthase activity was analyzed in the membrane fractions obtained from *Trichoderma* transformants and the control strain by incubation of 200 µg of membrane fraction with 100 mM potassium phosphate buffer pH 7.4, 5 mM MgCl_2_, 5 mM CHAPS, 10 mM DTT, 2 mM NADPH and 10 µM [^3^H]FPP in a total volume of 100 µL. Incubation was performed at 37 °C for 20 min, and then the reaction was stopped with 10 µL of 1 M EDTA pH 9.2, and 10 µL of unlabeled 0.5% squalene was added as a carrier. The reaction mixture was applied onto a Silica Gel 60 (Merck) thin-layer chromatography plate and developed in 5% (*v*/*v*) toluene in hexane. The radioactive zone containing squalene (R_f_ = 0.5) was scraped off and measured in a scintillation counter [31,32].

### 2.13. Protein Concentration Assay

Protein concentration was estimated according to Lowry et al. [33].

### 2.14. Identification and Quantification of Saccharides Bound to Secreted Proteins

The saccharides bound to proteins isolated from *T. atroviride* culture filtrates were assayed by the phenol-sulfuric acid procedure [34]. Secreted proteins were precipitated with two volumes of 96% ethanol, washed twice with 70% ethanol and dissolved in distilled water [35]. *O*-linked carbohydrates were removed from the protein pellet using mild alkaline hydrolysis according to Duk et al. [36] and analyzed in the supernatant, and then the concentration of *N*-linked carbohydrates was measured in the pellet. The calibration curve was prepared with D-mannose.

*O*- and *N*- linked carbohydrates obtained as above were hydrolyzed with 2 M TFA (trifluoroacetic acid) at 100 °C for 16 h. Monosaccharides were determined by high-performance anion-exchange chromatography using the Dionex ICS-3000 Ion Chromatography System with a Carbo Pac PA10 analytical column. Neutral sugars were eluted with 18 mM NaOH at 0.25 mL/min [37].

### 2.15. Cellulase Activity

The activity of cellulases was measured in the cultivation medium by incubation of 0.5 mL of carboxymethylcellulose (10 g/L) in 50 mM sodium citrate buffer (pH 5.0) at 50 °C for 2 h with 0.2 mL of culture filtrate. The reaction was stopped by boiling for 5 min. The amount of reducing sugars formed was determined by the method of Bernfeld [38] and estimated using a standard curve prepared with glucose. One unit of activity was defined as the amount of enzyme that liberated 1 µg of glucose from the substrate per 2 h.

### 2.16. Extraction and Purification of Polyisoprenoids

*Trichoderma* biomass was harvested by filtration, suspended in 60% KOH with 0.25% pyrogallol and hydrolyzed at 100 °C for 1 h. Lipophylic products were extracted with di-ethyl ether and evaporated to dryness, and then dissolved in hexane and applied onto a silica gel column equilibrated with hexane. The column was washed with 3% diethyl ether in hexane, and the polyisoprenoid fraction was eluted with 10% and 20% diethyl ether in hexane; the two eluates were pooled, evaporated to dryness, dissolved in isopropanol and subjected to HPLC/UV analysis as described earlier [39], with modifications. Briefly, a linear gradient of methanol: water (9:1, *v*/*v*) in methanol:isopropanol:hexane (2:1:1, *v*/*v*/*v*) was used for elution. Fractions containing polyisoprenoids (polyprenols and dolichols) were collected (based on comparison of their retention times with those of yeast dolichols used as external standards) and analyzed. For quantitative analysis, Dol13 was added to the biomass as an internal standard before hydrolysis. The polyprenol and dolichol standards were from the Collection of Polyprenols (IBB PAS).

### 2.17. Squalene Determination

Sterols were extracted according to the method proposed by Bernat et al. [40] with some modifications. First, 30 mg of lyophilized biomass was transferred into Eppendorf tubes containing glass beads, 0.66 mL of methanol and 0.33 mL of chloroform. The homogenization process using a ball mill (FastPrep-24, MP-Biomedicals) was carried out for 2 min. Next, the sample was centrifuged (2 min, 6000× *g*). The mixture was transferred to another Eppendorf tube. In order to facilitate the separation of two layers, 0.2 mL of H_2_O was added. The lower layer was collected and evaporated. Then, samples were diluted in 1 mL of methanol:chloroform (4:1, *v*/*v*). Sterols were measured using an Agilent 1200 HPLC system (Santa Clara, CA, USA) and a 3200 Q-TRAP mass spectrometer (Sciex, Framingham, MA, USA) equipped with the atmospheric pressure chemical ionization (APCI) source operating in the positive ionization mode. Then, 10 μL of the lipid extract was injected onto a Kinetex C18 column (50 mm × 2.1 mm, particle size: 5 μm; Phenomenex, Torrance, CA, USA) heated to 40 °C with a flow rate of 0.8 mL min^−1^. Water (A) and methanol (B) were applied as a mobile phase, both consisting of 5 mM ammonium formate. The solvent gradient was initiated at 40% B, increased to 100% B during 1 min and maintained at 100% B for 3 min before returning to the initial solvent composition over 2 min. The following instrumental settings were applied: curtain gas 25.0, ion spray voltage to 5500 nebulizer gas 50, auxiliary gas 50 and ion source temperature of 550 °C, entrance potential to 10.0. The monitored multiple reaction monitoring (MRM) pairs were 411.4–231.4, 411.4–163.3 for squalene. The data analysis was performed with the Analyst™ v1.5.2 software (Sciex, Framingham, MA, USA).

### 2.18. Plate Confrontation Assay

Mycelial disks (5 mm) of *T. atroviride RER2* transformants and *P. ultimum* were placed at opposite sides (7 cm apart) of agar plates with MM supplemented with 1% glucose. The plates were incubated at 28 °C, and photographs were taken after 3 days of incubation. Three replicates were prepared for each experiment and for each transformant.

### 2.19. Growth Inhibition of Fungal Pathogen by Agents Secreted by Trichoderma to Cultivation Medium

To determine its antagonistic activity, *Trichoderma* was grown on solid minimal medium with lactose covered with cellophane. Following the removal of the cellophane with *Trichoderma*, the effect of postculture medium containing secreted hydrolytic enzymes and metabolites on growth of *P. ultimum* was analyzed. A 5 mm mycelial disk of *P. ultimum* was placed in the center of the *Trichoderma*-pretreated plate. Plates were incubated at 28 °C for eight days, and the area of growth was measured. As a control, *P. ultimum* was cultivated on non-pretreated plates.

### 2.20. Statistical Analysis

For statistical analysis, we used the Student’s *t* test to compare the mean and standard deviation obtained for the mutants and the control strain. Standard significance level *p*-value of 0.05 was used.

## 3. Results

### 3.1. Expression of S. cerevisiae RER2 Gene in T. atroviride

The biocontrol *T. atroviride* strain P1 was transformed with the *RER2* gene from *S. cerevisiae*, and stable transformants were isolated, as detailed in Section 2 The presence of the yeast gene in the genome of the transformants was analyzed by PCR using RER2U (5′ CGGGATTCATGGAAACGGATAGTG GTATA 3′) and RER2L (5′ CGGATATCTTAATTCAACTTTTTTTCTTTC AAATC 3′) primers [18] and genomic DNA isolated from the transformants (Appendix A). As a positive and negative control, the PCR reactions were performed on the templates of DNA from *S. cerevisiae* and untransformed *T. atroviride*, respectively. Two transformants, RER30/11 and RER32/11, were obtained and subjected to further analysis. A copy number of *RER2* gene integrated into the *Trichoderma* genome was estimated by quantitative real-time PCR (qRT-PCR) analysis using the relative standard curve method [22,23,24]. It was established that transformants differ in the number of copies of the *RER2* gene integrated into their genome. Two copies were found in the RER30/11 strain, while only one copy was found in the RER32/11 transformant.

To quantify *RER2* expression in the transformed strains, mRNA was isolated, and reverse transcription qPCR analysis was performed. A significant expression of the *RER2* gene was observed in both transformants (Figure 2). The expression was tightly dependent on the number of *RER2* copies.

Furthermore, expression of the native *rer2* gene from *Trichoderma* was increased in the transformants compared to the control strain P1.

Thus, the PCR analyses showed that the *RER2* gene was integrated into the genome of the *T. atroviride* transformants and expressed in both transformants. Moreover, expression of the native *rer2* gene was slightly elevated, and expression of *nus1*, encoding subunit of the *cis*-prenyltransferase complex, was decreased in these strains compared to the control.

### 3.2. Biochemical Characterization of T. atroviride RER2 Transformants

To reach higher activity of *cis*-prenyltransferase, *RER2* gene from *S. cerevisiae* was used for expression in *T. atroviride* since activity of *cis*-prenyltransferase from yeast is several times higher compared to the homolog from *Trichoderma* [18,25,41].

Expression of the *RER2* gene increased the activity of *cis*-prenyltransferase in the membrane fraction obtained from the *RER2* transformants compared to the control strain P1. The higher increase, by 93%, was observed for the RER30/11 strain, while RER32/11 revealed a 30% increase (Figure 3). The higher activity of *cis*-prenyltransferase in the RER30/11 transformant is consistent with the higher expression of the *RER2* gene in this strain (Figure 2).

*Cis*-prenyltransferase uses farnesyl diphosphate (FPP) synthesized by FPP synthase as a substrate. The increased activity of *cis*-prenyltransferase in the transformants consumes more substrate, and this in turn could stimulate FPP synthase for more intensive production. Analysis of the FPP synthase activity confirmed this assumption: it was increased by 23 (RER32/11) and 29% (RER30/11) in the transformants compared to the control strain (Figure 3). FPP is also consumed by squalene synthase. Our study revealed that activity of squalene synthase was increased by 74 and 29% for RER30/11 and RER32/11, respectively, compared to the control strain (Figure 3).

The higher activities of *cis*-prenyltransferase and squalene synthase are reflected in a higher content of dolichols and squalene in the mycelia of the *RER2* transformants (Figure 4). Concentration of dolichols was 30% and 21% elevated in the RER30/11 and RER32/11 transformants, respectively, compared to the control strain. Concentration of squalene was also increased. A higher increase, by 52%, was observed for the RER30/11 strain, while RER32/11 revealed only a 5% increase, and the latter difference was not statistically significant. Increase in concentration of the both products, dolichols and squalene, is consistent with higher activity of the corresponding enzymes, *cis*-prenyltransferase and squalene synthase, respectively (Figure 3 and Figure 4).

Since *cis*-prenyltransferase produces dolichols and phosphorylated dolichols (Dol-P) serve as acceptors of carbohydrate residues for glycosylation, we examined the activity of dolichyl phosphate mannose (DPM) synthase producing the active form of mannose, DPM, for *O*- and *N*-glycosylation of proteins. Both transformants revealed higher activity of DPM synthase compared to the control (Figure 5). A higher increase, by 37%, was found in the membrane fraction of RER32/11, while in RER30/11, the activity was elevated only by 17%. We also analyzed the activity of *N*-acetylglucosamine transferase, the first enzyme in the synthesis of Dol-PP-oligosaccharide, used for *N*-glycosylation. Its activity was elevated by ca. 40% in both transformants compared to the control (Figure 5).

A higher production of DPM and DolPP-*N*-acetylglucosamine seemed likely to enhance the glycosylation of secretory proteins. In turn, it is hypothesized that the enhanced glycosylation of cellulases could influence their enzymatic activity.

To analyze the glycosylation of secretory proteins, *Trichoderma* was cultivated in minimal medium with lactose as a carbon source. Proteins were precipitated from culture medium after 72 or 96 h of cultivation, and the amount of *O*- and *N*-linked carbohydrates was determined.

A nearly 70% increase in the content of *O*-linked carbohydrates was found for both transformants compared to the control, while the amount of *N*-linked sugars was elevated more than fourfold (Figure 6). A qualitative analysis of the *O*- and *N*-linked polysaccharides by high-performance anion-exchange chromatography showed that mannose was the dominant sugar in both types of polycarbohydrates. A higher amount of glucose and mannose in the *N*-linked polycarbohydrates was found for RER30/11 strain, and it was, respectively, 7.4-fold and 3.7-fold that of the control. The *O*-linked carbohydrates of both transformed strains contained nearly the same amount of galactose as the control, while the amount of mannose was 1.75-fold and 2.1-fold higher for the RER32/11 and RER30/11 transformants, respectively (Figure 6).

### 3.3. Cellulolytic Activity of RER2-Expressing T. atroviride

The aim of this study was to elevate the secreted hydrolytic activity of *T. atroviride*.

Activity of cellulases was measured in the medium during cultivation.

Higher activity was found in the cultivation medium of RER30/11 (Figure 7). It was two-fold higher compared to the control strain. The activity registered in the medium of RER32/11 was much lower, but still it was 1.2-fold higher than that in the control.

The activity of cellulases reached the maximum after 96 h of cultivation.

The higher enzymatic activity of cellulases, secreted by the transformants was taken as an indicator for a pronounced inhibition of the growth of pathogenic Oomycete, particularly of *Pythium ultimum* containing a high proportion of cellulose in its cell wall. To prove this hypothesis, *Trichoderma* was cultivated on agar plates covered with cellophane, which was permeable to low-molecular-weight metabolites and hydrolases. After removing the cellophane together with the mycelia, the plant pathogen *P. ultimum* was cultivated on the pretreated plates. The growth of *P. ultimum* was inhibited by 70% on plates previously overgrown by the control strain, while on the plates pretreated by the transformants, the inhibition reached 89% (Figure 8).

In addition, we made a confrontation assay between the new *Trichoderma* strains and *P. ultimum* (Figure 9). *Trichoderma* and *Pythium* were cultivated on a single plate from separate inocula. We observed that *Pythium* was overgrown more rapidly by both transformants than by the control P1 strain and that the RER30/11 transformant was the most efficient.

## 4. Discussion

The aim of this study was to construct novel strains of *T. atroviride* with elevated hydrolytic capabilities. It is well known that during a direct attack, *Trichoderma* degrades the cell wall of a fungal pathogen with a variety of secreted extracellular enzymes [42]. Based on this knowledge, we decided to increase the hydrolytic activity of the new strains. Our previous studies have shown an interdependence between the level of glycosylation of secretory proteins and their hydrolytic activity. Furthermore, we observed that improved processing of secretory proteins in the endoplasmic reticulum elevated their secretion [12,13,31]. We improved glycosylation by increasing synthesis of dolichols by *cis*-prenyltransferase.

The basal activity of *cis*-prenyltransferase in *T. atroviride* was much lower than that in *S. cerevisiae* [18,25,41]. Therefore, the use of *RER2* gene from *S. cerevisiae* for expression in *Trichoderma* was very promising for achieving high activity of *cis*-prenyltransferase in the transformed strains. In addition, Rer2 protein from yeast had to form a functional complex with Nus1p from *Trichoderma*, giving increased activity of *cis*-prenyltransferase in the *RER2* transformants of *T. reesei* [18]. It was shown that Nus1protein together with Rer2p were required for *cis*-prenyltransferase activity [43].

The glycosylation of secretory proteins has been postulated to protect them against proteolysis, and, moreover, it has been shown to directly affect the conformation and stability of the secreted enzymes [4,6,9,10,11]. In addition, it has been reported that *O*-glycosylation is necessary for secretion of hydrolases by *Trichoderma* [44]. CBHI (cellobiohydrolase I), the main protein secreted by *Trichoderma*, consists of a catalytic domain and a cellulose binding module joined by a highly *O*-glycosylated linker peptide. The catalytic domain has four *N*-linked motifs, three of which are glycosylated. The type and extent of glycan on the catalytic domain are influenced by both the strain and culture condition [45]. In addition, the amount of carbohydrates bound to secretory proteins varies during cultivation [18].

In the present report, the highest glycosylation of secretory proteins appeared after 96 h of cultivation of the *RER2*-transformed strains and correlated with their highest hydrolytic activity.

The hydrolases secreted by *Trichoderma* to the medium strongly inhibited the growth of *P. ultimum,* an *Oomycetes* whose cell wall contains mainly 1,3-β glucans, 1,6-β-glucans and 1,4-β-glucans (cellulose). Chitin, the major constituent of the “typical” fungal cell wall, has been detected in small amounts in few *Oomycetes* [46]. Since the cellulolytic activity secreted by the *Trichoderma* transformants was significantly elevated compared to the nontransformed strain, it resulted in substantially higher antimicrobial activity of these strains against *P. ultimum*.

In summary, these results show that posttranslational modification of secretory proteins is a promising target for improving the effectiveness of the direct attack of *Trichoderma* on fungal pathogens. This study and our previous results confirm that simple genetic manipulations to enhance the activity of glycosylation-related enzymes, such as DPM synthase, guanylyltransferase, farnesyl pyrophosphate synthase and *cis*-prenyltransferase, produce pleiotropic effects that ultimately result in a higher activity and/or secretion of hydrolytic enzymes [12,13,14,31,47]. These results are in contrast to more direct attempts to enhance the hydrolytic activity of *Trichoderma* by overexpression of genes coding for a single lytic enzyme [48,49]. The biocontrol effects of such manipulations were rarely significant, probably because polymers such as cellulose, glucan and chitin require several hydrolytic enzymes for degradation. By improving protein glycosylation, we obtained a more general effect not limited to a single enzyme.

## Figures and Tables

**Figure 1 jof-09-00038-f001:**
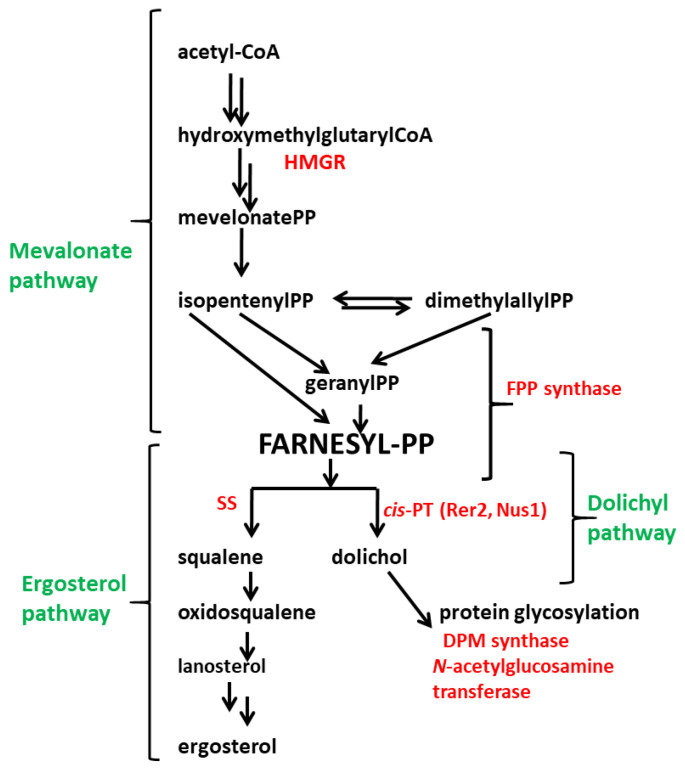
Metabolic pathways analyzed in this study. Abbreviations presented in the figure: HMGR—3 hydroxy-3-methylglutaryl–CoA reductase (regulatory enzyme of the mevalonate pathway); FPP synthase—farnesyl diphosphate synthase; SS—squalene synthase; *cis*-PT (Rer2)—*cis*-prenyltransferase; Nus1-*cis*-PT complex subunit; DPM synthase—dolichyl phosphate mannose synthase. The names of the pathways are marked in green, and the names of the enzymes analyzed in this study in red.

**Figure 2 jof-09-00038-f002:**
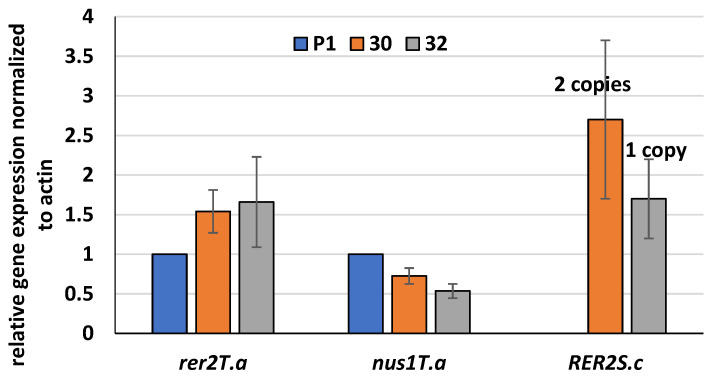
Transcript levels of *RER2* genes from *S. cerevisiae* (*RER2S.c*) and *T. atroviride* (*rer2T.a*) and *nus1* subunit of *cis*-prenyltransferase (*nus1T.a*) determined by RT-qPCR in the *RER2* transformed *T. atroviride*. *Trichoderma* strains (30—RER30/11; 32—RER32/11) and control strain P1 were grown for 144 h in PDB medium, RNA was extracted and cDNA synthesized. qPCR reactions were performed using a LightCycler 96 instrument. The amplification and melting curve data were collected and analyzed using the LightCycler96^®^ software 1.0. Data were obtained from three independent experiments, each determined in triplicate. Expression of the native *rer2* gene in the transformed strains was normalized to *rer2* expression in the control strain P1.

**Figure 3 jof-09-00038-f003:**
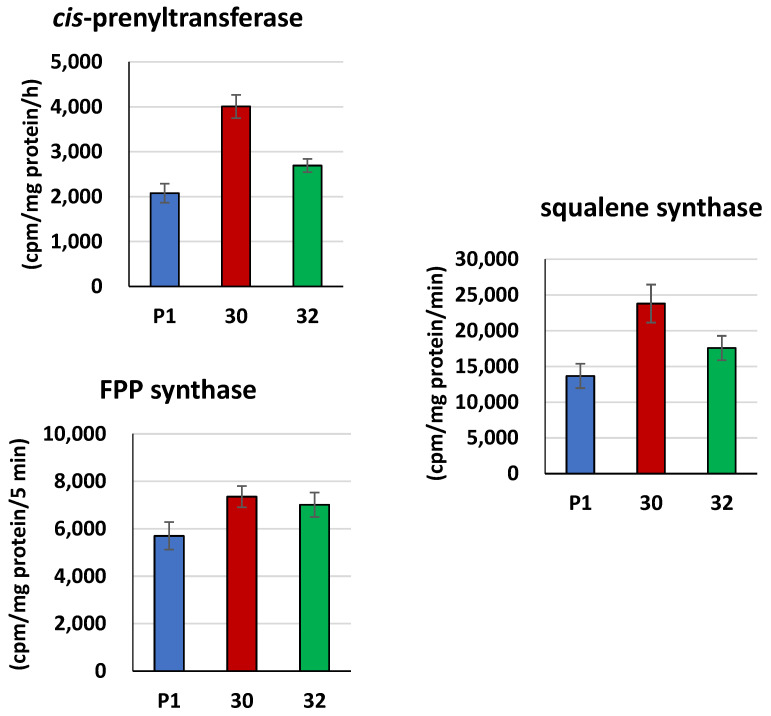
Activities of *cis*-prenyltransferase and squalene synthase in membrane fraction or FPP synthase in cell-free extracts from *RER2* transformed *T. atroviride*. *Trichoderma* strains (30—RER30/11; 32—RER32/11) and control strain P1 were processed and analyzed for enzymatic activity, as described in Section 2. Data are presented as mean ± standard deviation from six independent experiments, each determined in triplicate. Differences are statistically significant (*p* < 0.05; *t* test).

**Figure 4 jof-09-00038-f004:**
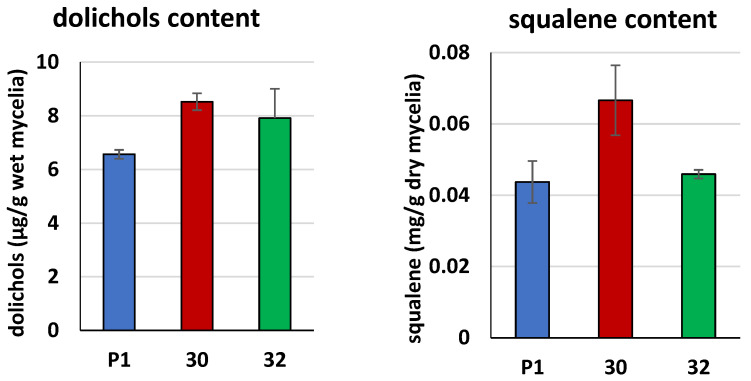
Concentration of dolichols and squalene in the mycelia from *RER2* transformed *T. atroviride. Trichoderma* strains (30—RER30/11; 32—RER32/11) and control strain P1 were processed and analyzed for the concentration of products synthesized by *cis*-prenyltransferase and squalene synthase, dolichols and squalene, respectively. Data are presented as mean ± standard deviation from three independent experiments, each determined in triplicate. Differences are statistically significant (*p* < 0.05; *t* test).

**Figure 5 jof-09-00038-f005:**
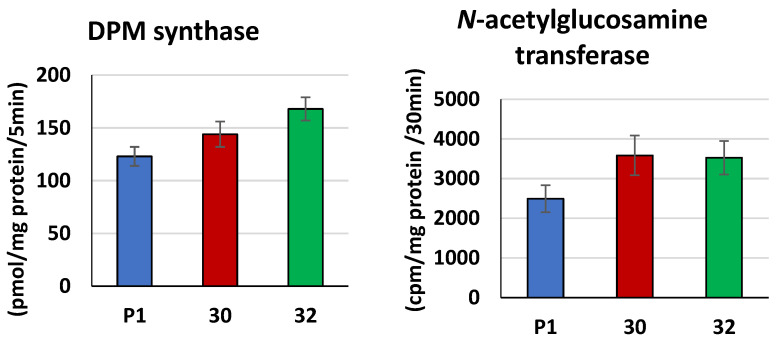
Activities of DPM synthase and *N*-acetylglucosamine transferase in membrane fraction from *RER2* transformed *T. atroviride. Trichoderma* strains (30—RER30/11; 32—RER32/11) and control strain P1 were processed and analyzed for enzymatic activity as described in Section 2. Data are presented as mean ± standard deviation from six (DPM synthase) or four (*N*-acetylglucosamine transferase) independent experiments, each determined in triplicate. Differences are statistically significant (*p* < 0.05; *t* test).

**Figure 6 jof-09-00038-f006:**
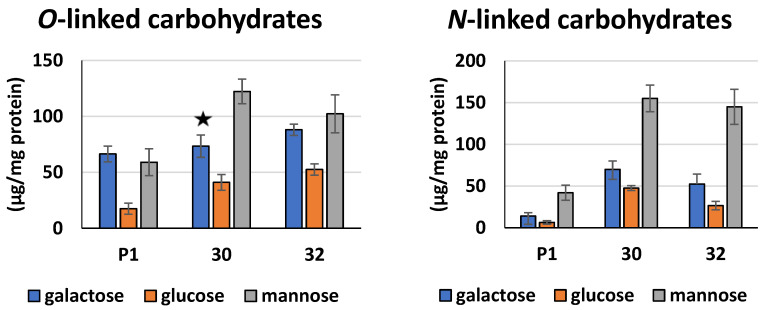
Content of *O*- and *N*-linked carbohydrates in proteins secreted by *RER2* transformed *T. atroviride* after 96 h of cultivation. *O*- and *N*-linked carbohydrates were released selectively from proteins secreted by transformants (30—RER30/11; 32—RER32/11) and control strain P1 and determined by high-performance anion-exchange chromatography. Data are presented as mean ± standard deviation from three independent experiments. ★—differences statistically insignificant (*p* < 0.05; *t* test).

**Figure 7 jof-09-00038-f007:**
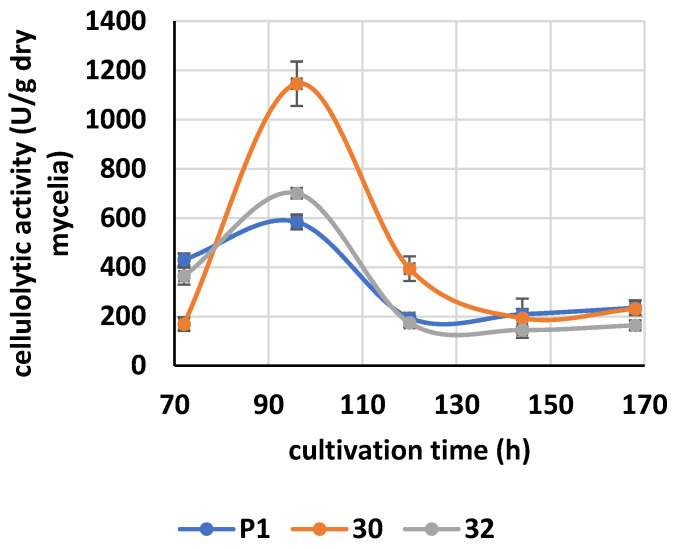
Activities of cellulases in cultivation medium from *RER2*-transformed *T. atroviride.* Concentration of reducing sugars released from carboxymethylcellulose by cellulases secreted into cultivation medium by the transformants (30—RER30/11; 32—RER32/11) and the parental strain P1 were measured. Data are presented as mean ± standard deviation from six independent experiments. Differences are statistically significant (*p* < 0.05; *t* test).

**Figure 8 jof-09-00038-f008:**
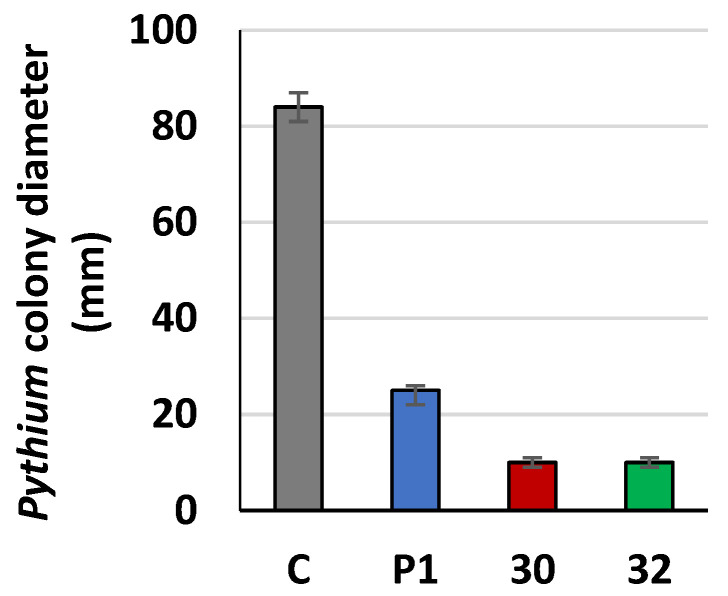
Growth inhibition of *P. ultimum* cultivated on plates pretreated with *RER2* transformed *T. atroviride. Trichoderma* strains (30—RER30/11; 32—RER32/11) and control P1strain were cultivated for three days on MM plates covered with cellophane and then removed with the cellophane. *P. ultimum* was inoculated on the pretreated plates, and its rate of growth was determined as colony diameter after two days. As a control (C), *P. ultimum* was cultivated on non-pretreated plates. Data are presented as mean ± standard deviation from six independent experiments. Differences are statistically significant (*p* < 0.05; *t* test).

**Figure 9 jof-09-00038-f009:**
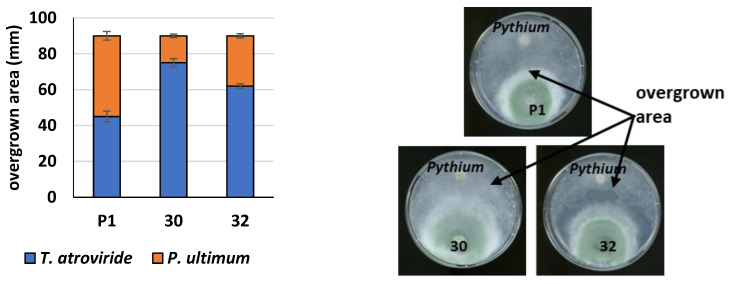
Plate confrontation assay of *T. atroviride* against *P. ultimum*. Mycelial disks of RER30/11 (30), RER32/11 (32) transformants and the control strain (P1) and *P. ultimum* were placed at opposite sides of agar plates and incubated at 28 °C. Pictures were taken three days after inoculation. Arrows mark the overgrown zone between the two fungi. The overgrown zones were measured, and results are presented as mean ± standard deviation from three separate plates. Differences are statistically significant (*p* < 0.05; *t* test).

## Data Availability

Not applicable.

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
