# Peer review of "Expression of Saccharomyces cerevisiae RER2 Gene Encoding Cis-Prenyltransferase in Trichoderma atroviride Increases the Activity of Secretory Hydrolases and Enhances Antimicrobial Features"

_jof, 2022, doi:10.3390/jof9010038_

Round 1
Reviewer 1 Report
In this paper, the authors created and characterized new Trichoderma atroviride strains with elevated antifungal activity by expressing the RER2 gene from Saccharomyces cerevisiae. Cellulases secreted by the transformants revealed significantly higher levels or activities compared to the control strain, resulting in that Trichoderma strains were more effective against the plant pathogens Pythium ultimum. There are some questions that I think help improve this manuscript.
1. In Figure 1, what the double arrows mean? Also, what the words with red and green color represent?
2. In Figure 2, please address the statistical analysis, as well as Figure 3-9. What the nus1T.a means in Fig. 2?
3. In Introduction section, the RER2 should be addressed.
4. The reasonable explanation that why DPM synthase and O-linked carbohydrates in 30 seem to be lower than those in 32 should be addressed.
5. The definition of unit (U) of CMCase activity is lack.
6. Some language expressions need to be improved.
Author Response
Reviewer 1
In this paper, the authors created and characterized new Trichoderma atroviride strains with elevated antifungal activity by expressing the RER2 gene from Saccharomyces cerevisiae. Cellulases secreted by the transformants revealed significantly higher levels or activities compared to the control strain, resulting in that Trichoderma strains were more effective against the plant pathogens Pythium ultimum. There are some questions that I think help improve this manuscript.
- In Figure 1, what the double arrows mean? Also, what the words with red and green color represent?
Double arrows mean that there is not one reaction but two or more steps to get the next product. The names of the pathways are marked in green, and the names of the enzymes analyzed in this study in red. In addition, we added the position of HMGR as the main regulatory enzyme of the mevalonate pathway. (color of HMGR was changed from green to red) – information added to the description of Figure 1
- In Figure 2, please address the statistical analysis, as well as Figure 3-9. What the nus1T.a means in Fig. 2?
For statistical analysis we used the Student’s t-test to compare the mean and standard deviation obtained for our mutants and the control strain. Standard significance level p-value of 0.05 was used. – information added to the Methods 4.20
nus1T.a is an additional subunit of cis-prenyltransferase. – information added to the Fig. 2 description
- In Introduction section, the RER2 should be addressed.
Text added line 71-75
- The reasonable explanation that why DPM synthase and O-linked carbohydrates in 30 seem to be lower than those in 32 should be addressed.
Mannosyl residues coming from DPM are also present in the N-linked carbohydrates. It is impossible to count them precisely. The amount of carbohydrates bound to the secretory proteins is changeable during time of cultivation. We observed that the amount of carbohydrates decreased gradually during cultivation. O-linked carbohydrates were more stable [17] – Perlinska-Lenart et al. 2006
- The definition of unit (U) of CMCase activity is lack.
One unit of activity was defined as the amount of enzyme that liberated 1 µg of glucose from the substrate per 2 h. –added to the Method 4.15
- Some language expressions need to be improved.
Language improved
Reviewer 2 Report
This manuscript reported the heterologous overexpression of a cis-prenyltransferase, namely RER2, from Saccharomyces cerevisiae in Trichoderma atroviride. The gene-modified strain enhanced the activities of dolichol-dependent enzymes, DPM synthase, N-acetyl-glucosamine transferase, and cellulase activity and stimulated glycosylation of secretory proteins.
Authors demonstrated that “cellulases secreted by the transformants revealed significantly higher levels or activities compared to the control strain. Consequently, the resulting Trichoderma strains were more effective against the plant pathogens Pythium ultimum.”
However, the results obtained did not support this conclusion. Enhanced cellulase activity could result from enhanced secretion of the enzyme or enhanced specific activity of the enzyme. Thus, experiments to verify the cellulase gene expression at transcription and expression levels should be done to make a solid conclusion.
On the other hand, the conclusion that enhanced cellulase activity resulted in increased antimicrobial activity is also not solid. Authors need to purify the target cellulase and test its glycosylation state, biochemical properties, and antimicrobial activity.
Author Response
file enclosed

Reviewer 3 Report
The manuscript deals with the hetereologous overexpression of a glycosylation pathway gene in Trichoderma atroviride in order to increase its antifungal properties. The manuscript is well written and clear and it can add knowledge to the fungal scientific community.
Some points should be adressed:
Was there a RER2 homologous in T. atroviride genome? If so, why not overexpressing the homologous gene or using the gene from other filamentous fungi?
Why was lactose used as coarbon source for Trichoderma cultivation? Can the efficiency also be achived with glucose only?
The introcution or discussion should bring more discussion/description of the expected glycans. What are they compositions?
Author Response
file enclosed

Round 2
Reviewer 1 Report
No comments
Author Response
I thank the Reviewer for fruitful cooperation. I hope all language corrections have been made.
Reviewer 2 Report
Authors stated that "RER32/11 and the control P1 secret the same amount of proteins and RER30/11 secrets more. On the other hand, carbohydrates bound to the secretory proteins and cellulases activity are counted per mg of proteins."
Did the fungus only secret cellulase? If not, this conclusion is still not solid. Authors need to provide a SDS-PAGE of extracellular proteins.
Author Response
file added
